# Genome-Wide Association Studies for Sex Determination and Cross-Compatibility in Water Yam (*Dioscorea alata* L.)

**DOI:** 10.3390/plants10071412

**Published:** 2021-07-10

**Authors:** Jean M. Mondo, Paterne A. Agre, Robert Asiedu, Malachy O. Akoroda, Asrat Asfaw

**Affiliations:** 1International Institute of Tropical Agriculture (IITA), Ibadan 5320, Nigeria; m.mubalama@cgiar.org (J.M.M.); r.asiedu@cgiar.org (R.A.); a.amele@cgiar.org (A.A.); 2Institute of Life and Earth Sciences, Pan African University, University of Ibadan, Ibadan 200284, Nigeria; 3Department of Crop Production, Université Evangélique en Afrique (UEA), Bukavu 3323, Democratic Republic of the Congo; 4Department of Agronomy, University of Ibadan, Ibadan 200284, Nigeria; malachyoakoroda@gmail.com

**Keywords:** dioecy, sex determination, cross-pollination success, marker development, *Dioscorea alata*

## Abstract

Yam (*Dioscorea* spp.) species are predominantly dioecious, with male and female flowers borne on separate individuals. Cross-pollination is, therefore, essential for gene flow among and within yam species to achieve breeding objectives. Understanding genetic mechanisms underlying sex determination and cross-compatibility is crucial for planning a successful hybridization program. This study used the genome-wide association study (GWAS) approach for identifying genomic regions linked to sex and cross-compatibility in water yam (*Dioscorea alata* L.). We identified 54 markers linked to flower sex determination, among which 53 markers were on chromosome 6 and one on chromosome 11. Our result ascertained that *D. alata* is characterized by the male heterogametic sex determination system (XX/XY). The cross-compatibility indices, average crossability rate (ACR) and percentage high crossability (PHC), were controlled by loci on chromosomes 1, 6 and 17. Of the significant loci, SNPs located on chromosomes 1 and 17 were the most promising for ACR and PHC, respectively, and should be validated for use in *D. alata* hybridization activities to predict cross-compatibility success. A total of 61 putative gene/protein families with direct or indirect influence on plant reproduction were annotated in chromosomic regions controlling the target traits. This study provides valuable insights into the genetic control of *D. alata* sexual reproduction. It opens an avenue for developing genomic tools for predicting hybridization success in water yam breeding programs.

## 1. Introduction

Yam (*Dioscorea* spp.) is an important food and cash crop in tropical and subtropical areas [1]. It is extensively produced (~93% of world production) in the African yam belt, a six-country region from Cameroon to Côte d’Ivoire, where it plays significant economic, sociocultural, and religious roles among ethnic groups [2]. *Dioscorea alata*, commonly referred to as water, winged or greater yam, is the most widely distributed and the second-most-produced yam species after *D. rotundata* worldwide [3]. The popularity of *D. alata* stems from its high yield potential (even under low soil fertility), ease of propagation, competition with weeds (early vigor) and tuber storability [4,5]. Yam yield has, however, remained low over time because of several biotic (diseases and pests), abiotic (drought, low soil fertility, etc.), and agronomic constraints [6,7]. Developing resistant/tolerant varieties coupled with a robust seed delivery system could be an effective means of raising yields of predominantly resource-poor farmers characterized by low use of external farm inputs. The variety development process requires a thorough understanding of the crop’s reproductive mechanisms.

Yam is a monocotyledonous herbaceous vine plant that reproduces vegetatively through tubers or vines or sexually through botanical seeds [8]. Yam is mainly dioecious with male and female flowers on separate plants, although monoecious plants with both male and female flowers on the same individuals exist [3,9,10,11]. Flowering and flower sex in plants are most strongly determined by genotype, although environmental, hormonal and epigenetic cues, to some extent, bear influence. The genetic mechanisms range from a single locus to sex chromosomes bearing several linked loci required for sex determination [12,13]. Dioecy in plants is inherited via three sex chromosome systems: XX/XY, XX/X0 and WZ/ZZ, such that XX or WZ determines female sex phenotype and XY, X0 or ZZ the male sex phenotype [12,13,14,15]. Most of the studied yam species, such as *D. alata* [3,5], *D. floribunda* [16], and *D. tokoro* [17] are characterized by the male heterogametic (XX/XY) sex-determination system. However, *D. rotundata* [9] and *D. deltoidea* [18] possess a female heterogametic (ZZ/ZW) sex-determination system. It is noteworthy that the *D. alata* species used for this study is strictly dioecious, with no monoecy-reported cases [3].

Given the dioecious nature of *D. alata* plants, sex identification at the seedling stage is crucial for genetic improvement through breeding. As in most plant species, sexual/gender dimorphism (apparent morphological, physiological and life-history trait differences among females and males) in *D. alata* is negligible at the vegetative stage. Hence, male and female individuals may not be reliably identified before flowering by visual observations [19]. The use of molecular markers is the most reliable strategy in discriminating yam clones for flower sex at early growth stages [10,19].

Some markers linked to sex chromosomal regions have been identified for both popular yam species (*D. alata* and *D. rotundata*) [3,5,9]. Tamiru et al. [9] identified a female-specific chromosomic region on the pseudo-chromosome 11 of *D. rotundata*. They developed a single nucleotide polymorphism, (SNP) marker sp16, for yam plant sex identification at the early seedling stage. However, the sp16 marker only predicts the likelihood of femininity and may not be transferable to other species. Tamiru et al. [9] also identified a DNA marker, sp1, linked to the putative Z-linked region predicting maleness. Using these markers to predict sex at earlier growth phases among *D. rotundata* accessions has been reported [10,20]. However, these markers’ prediction accuracy is not always perfect in yam sex identification at the seedling stage. A *D. alata* sex determination region was mapped on chromosome 6 and a kompetitive allele-specific PCR (KASP) marker for accurate cultivar sex estimation was developed [3,5]. However, no report exists on its practical application for identifying flower sex at the seedling stage in *D. alata*. Sex determination in yam plants could be controlled by more than one locus [20], and thus, identifying more sex markers is encouraged. Besides, the instability of the sexual phenotype across generations and environments is another indication that sex expression in yam is a polygenic trait [21].

Another major issue during yam hybridization activities is the low cross-compatibility rates among cultivars (~23 and 31% for *D. rotundata* and *D. alata*, respectively) [22]. However, efforts to establish an efficient method to unravel the genetic basis of the cross-compatibility in yam are very limited. An accurate method for early detection of seedling sex and compatible fertile parents prior to designing cross-combinations would be helpful to improve cross-pollination success in yam breeding.

Most of the previous studies on yam flowering and sex determination used bi-parental populations, with the probability that findings could have been related to parental specificity. The use of the genome-wide association (GWAS) approach could be helpful to ascertain results from previous studies and to identify more genomic regions controlling target traits. Guo et al. [23] showed the potential of association mapping (GWAS) for connecting genomics and phenomics for natural outcrossing in rice. Several other studies have successfully applied GWAS for flowering time and sex determination studies [21,24,25]. In this study, we used the Diversity Array Technology (DArT) for sequencing, which is a robust and low-cost high-throughput open platform method for DNA polymorphism analysis [26]. It provides high call rates and scoring reproducibility compared to other sequencing techniques. Besides, DArT has been successfully used in water yam research to explore genetic diversity, evolution, population structure and identification of loci linked to disease resistance and tuber quality traits [27,28,29].

The objective of this study was to investigate, using the GWAS approach, the genomic regions linked to sex determination and cross-compatibility for improving the pollination efficiency in water yam hybridization activities.

## 2. Results

### 2.1. Sex and Cross-Compatibility Indices of D. alata Clones Used for GWAS Analyses

This study used 2010–2020 historical pollination data of 74 *D. alata* genotypes to investigate genomic regions controlling plant sex and cross-pollination success rate at the International Institute of Tropical Agriculture (IITA) breeding sites in Nigeria. Phenotypic data on flower sex, average crossability rate (ACR) and percentage high crossability (PHC) are presented in Table 1. Of the 74 genotypes, 33 were female and 41 were male flowering phenotypes. The ACR of the studied genotypes ranged from 1.59% for TDa9801176 to 91.04% for TDa1401253, with a mean of 49.4%. The PHC ranged from 0 to 100%. Among parental clones, TDa9900240 and TDa0200012 were the most used female and male parents, respectively, involved in over 40 cross-combinations.

### 2.2. Chromosomic Regions Linked to D. alata Sex Determination and Cross-Compatibility

The GWAS scan identified 54 SNP markers associated with variation for flower sex; 53 of these markers were located on chromosome 6 while one was on chromosome 11 (Table 2, Figure 1a). Of the total SNP markers associated with plant sex, the minor allele frequencies (MAF) ranged from 0.13 (Chr6_837364 and Chr6_843525) to 0.43 (Chr6_3465 and Chr6_53812). The total phenotypic variance explained (PVE) by inventoried SNP markers was high (49–86%). The marker effects ranged from −1.92 to 1.77. The logarithm of odd (LOD)-scores varied from 4.47 to 9.69 for sex markers (Table 2).

Three SNP markers distributed on three chromosomes (Figure 2a, Table 2) were identified as responsible for the genotype’s average crossability rate (ACR). Chr6_3161 is located at 3 kilo-base pairs (kbp) on chromosome 6 while the SNP Chr1_215056 on chromosome 1 is located at 21 kbp and Chr17_9492 on chromosome 17 at 9 kbp. PVE ranging from 32 to 35% was observed, with minor allele frequencies of 0.25–0.35, and the marker effects were from −20.47 to 14.02 (Table 2).

Two markers were found for the percentage high crossability (PHC) on chromosomes 1 and 6 (Figure 3a). The marker Chr1_215056 was from chromosome 1, at the physical position of 215 kbp, it explained 29% of the phenotypic variance, had a marker effect of −43.11 and a LOD-score of 4.01. This marker’s MAF was 0.03. On the other hand, the marker Chr6_3227 was retrieved at the position 3 kbp on chromosome 6. Its MAF was 0.26, and it explained 29% of the phenotypic variance. The marker effect and LOD-score were −27.36 and 4.04, respectively (Table 2).

The quantile–quantile (Q–Q) plots generated by plotting the negative logarithms (−log10) of the *p*-values against their expected *p*-values showed appropriateness of the GWAS model for all the three traits. There was an inflection between observed and expected values for target traits, thus supporting association between the phenotype and markers (Figure 1b, Figure 2b and Figure 3b).

### 2.3. Analysis of the Sex Determination System

The haplotype view of markers associated with plant sex in female and male plants of *D. alata* showed that the sex is controlled by the male parent (XY) since the females were 95.9% homozygous (XX) for markers linked to sex determination (Figure 4, Appendix A). In contrast, markers linked with plant sex displayed 84.96% heterozygosity in the male genotype population (Figure 5, Appendix A).

### 2.4. Haplotype Segregation for ACR and PHC

The haplotype segregation showed that among the three markers identified as associated with ACR, Chr17_9492 was the most promising in discriminating genotypes for pollination success (*p* < 0.05). Of the three variants (TT, CT and CC), the variant CC was associated with low ACR (Figure 6, Table 3). On the other hand, CT and TT were identified as predictors of genotypes with high ACR. The other two SNP markers for ACR showed no significant effects among the different variants. The marker Chr1_215056 allowed discrimination for the PHC: the haplotype AA was associated with high PHC, while the haplotype AG controlled low PHC (Figure 7, Table 3).

### 2.5. Putative Gene Annotation Linked to Flower Sex and Cross-Pollination

Inventoried gene or protein families with any association to plant flowering and reproduction are presented in Appendix A. We identified four gene/protein families in chromosomic regions associated with ACR: Homeobox domain, Helix-turn-helix motif, NAC domain and Zinc finger CCHC-type protein. Twelve gene families previously reported for their involvement in plant reproduction in other crops were found in chromosomic regions associated with the PHC. Of these, WD40 repeat G-protein, ubiquitin-protein ligase SINA like, Seven-in-absentia protein (TRAF-like domain), P-loop containing nucleoside triphosphate hydrolase and Proteasome component (PCI) domain are the most promising candidates. On the other hand, we identified 45 different gene/protein families with links to plant reproduction in chromosomic regions controlling plant sex determination. Among them, the most promising candidates were Zinc finger (RING/FYVE/PHD-type), Glycosyltransferase AER61, NAD(P)H-quinone oxidoreductase subunit L, GLABROUS1 enhancer-binding protein (GeBP) and GeBP-like proteins, Auxin efflux carrier, Ribosomal protein PSRP-3/Ycf65, MADS-box, RNA recognition motif domain, Basic-leucine zipper domain, Myb/SANT-like domain, Aldolase-type TIM barrel, NAD(P)-binding domain, Zinc finger (Rad18-type putative), Tify, ABC transporter-like, Homeodomain-like, Prohibitin, Initiation factor eIF-4 gamma (MA3) and HD-ZIP protein (N-terminal).

## 3. Discussion

### 3.1. Genomic Regions Controlling Sex Determination, ACR and PHC Are on the Same Chromosomes

An accurate method for early detection of seedling sex and compatible fertile parents prior to designing crosses would be helpful to improve cross-pollination success in yam breeding. We used GWAS method to investigate genomic regions controlling sex determination, ACR and PHC in *D. alata*. A total of 54 markers were identified for sex determination, 53 of these were mapped on chromosome 6 and one on chromosome 11. Besides, the gene annotation showed many gene/protein families previously involved in flowering and reproduction in other crop species on those chromosomes, especially chromosome 6. Our findings agree with previous sex determination and flowering behavior studies on *D. alata*. Previous reports demonstrated an involvement of chromosome 6, and secondary chromosomes 1 and 11, on *D. alata* flowering and sexual reproduction [3,5]. As hypothesized by Denadi et al. [20], *Dioscorea* spp. flowering and sex might be controlled by several genes as our results also supported. For instance, we have identified up to 54 SNP markers on chromosomes 6 and 11 for the flower sex determination. This implies that additional SNP markers should be developed for sex identification and deployed in yam breeding programs to complement efforts by Cormier et al. [5].

This study investigated for the first time the association between chromosomic regions and successful pollination rates (represented in this work by two indices: ACR and PHC) in yam. The GWAS output showed that the ACR was controlled by chromosomes 1, 6 and 17, while the PHC was associated with genes on chromosomes 1 and 6. We can, thus, conclude that chromosomes 1 and 6 are the major contributors to *D. alata* flower sex and cross-pollination success. Being controlled by the same chromosomes, there might be a probable correlation between flowering ability, sex determination and pollination success in water yam. If the correlation is confirmed, it would be possible to select for these traits simultaneously. Although not associating flower sex with chromosome 1, Cormier et al. [3] showed that this chromosome is involved in *D. alata* flowering ability. Other studies also found a relationship between genes controlling reproduction traits such as flowering time, behavior and sex in plants [21].

Of the identified markers, Chr17_9492 and Chr1_215056 were the most promising for ACR and PHC predictions, respectively. This work, therefore, opens an avenue for improving hybridization practices in water yam by providing molecular markers for sex determination (crucial for effective hybridization plans in dioecious plants) and the crossability potential of parents to be involved in breeding programs.

### 3.2. Dioscorea alata Flower Sex Is Controlled by a Male Heterogametic System

The haplotype analysis showed that sex in *D. alata* is determined by the male parent. The females were at ~96% homozygous for markers linked to sex determination while males were ~85% heterozygous for this trait. Our results, using GWAS approach, Diversity Arrays Technology (DArT) and a core collection from West Africa, supported previous reports on *D. alata* sex determination which showed that this species is characterized by a male heterogametic sex determination system [3,5]. In such a system, XX determines female sex phenotype and XY the male sex phenotype [3,5]. It is noteworthy that previous studies on *D. alata* sex determination used either bi-parental populations [5] or GWAS with genotyping-by sequencing (GBS) and a core collection from the French West Indies (Guadeloupe) in Latin America [3]. Besides, Cormier et al. [3] aligned raw sequencing reads on the *D. rotundata* reference genome v1 [9] to detect SNPs, while our study used the newly released *D. alata* reference genome [29]. Using a different approach and plant material, our study is, therefore, strengthening conclusions on *D. alata* sex determination as reported by previous studies. Efforts should be concentrated on the SNP markers which displayed 100% homozygosity for female genotypes, while markers with 100% heterozygosity record should be selected for future studies, such as marker conversion into KASP-PCR, validation and deployment in the breeding program for marker-assisted selection.

As also reported by Cormier et al. [3], we observed a certain level of mismatch between the genetic information and the sex phenotype of some *D. alata* cultivars, such that a genotype with male haplotypes could display a female phenotype and vice versa. These authors hypothesized that the ploidy level could possibly explain that mismatch. They argued that the polyploidy leads to major changes in gene regulation and expression, as also supported by Chen [30]. Therefore, efforts are necessary to elucidate the extent of the influence of ploidy level on flower sex prediction in *D. alata*.

The presence of gene/protein families regulating hormones such as gibberellins, auxins, ethylene and cytokinins in the sex-determining regions (Appendix A) could confirm the crucial roles played by these phytohormones for sex determination in dioecious and monoecious plants. Generally, auxins and ethylene have feminizing effects, whereas cytokinins and gibberellins have masculinizing effects [15,21,31,32,33]. A better understanding of the hormone balance for a sex phenotype display could facilitate manipulation of flowering behaviors and sex ratios in *D. alata*.

It is noteworthy that the equilibrium sex ratio of 1:1 expected from the Fisherian theory is seldom respected in *Dioscorea* species as there is a significant male bias [8,11,34]. The frequent occurrence of male-biased sex ratios in the plant has been associated with three factors: (i) greater female than male reproductive expenditure, (ii) greater sensitivity of females to stress and (iii) spatial segregation of the sexes as a result of resource gradients. Thus, the high reproductive investment required makes females more sensitive to internal (genetic) and exogenous (ecological) conditions affecting plant reproductive activities [12,19,35].

### 3.3. Gene Annotation Showed the Presence of Gene/Protein Families with Links to Plant Sex and Cross-Pollination

We identified a total of 61 gene/protein families with links to plant flowering and reproduction. Their specific functions, crops in which they were reported and corresponding references are provided in Appendix A.

Four gene/protein families were found on chromosomic regions associated with ACR: Homeobox domain (WOX genes); Helix-turn-helix motif (LEAFY (LFY) protein); NAC domain (NAC TFs, MtNAM); and Zinc finger CCHC-type protein (Mt-Zn-CCHC gene). Previous reports have demonstrated the role of these families in either floral morphology, flowering initiation, differentiation of egg cells and zygotes, seed pod formation, translocation toward fruit under stress conditions, regulation of floral organ identity or seed size in *Monotropa hypopitys* [36], *Arabidopsis thaliana* [37], *Citrullus lanatus* [38] or *Medicago truncatula* [39,40].

Of the 12 gene/protein families found in the PHC-controlling chromosomes (Appendix A), the most determinant were the WD40 repeat G-protein (GTS1, [41]); ubiquitin-protein ligase SINA like (UBP12, UBP13, UBP14, UBP15, [42]); Seven-in-absentia protein (TRAMGaP, At5g26290, [43]); P-loop containing nucleoside triphosphate hydrolase (Sin-2, [44]); and Proteasome component (PCI) domain (RPN5, [45]).

This study identified up to 45 different gene/protein families with links to plant reproduction in chromosomic regions controlling plant sex determination (Appendix A). The most popular gene/protein families with direct or indirect influence on sex determination or with differential expression among sexes or those regulating hormones linked to sex ratio bias in other plant species are listed below. These are: Zinc finger, RING/FYVE/PHD-type [46]; Glycosyltransferase AER61 [47]; NAD(P)H-quinone oxidoreductase subunit L [48]; GLABROUS1 enhancer-binding protein [49]; Auxin efflux carrier [50]; Ribosomal protein PSRP-3/Ycf65 [15]; MADS-box [51,52,53]; Myb/SANT-like domain [54]; Aldolase-type TIM barrel [55]; NAD(P)-binding domain [56]; Zinc finger, Rad18-type putative [57]; Tify [50,58]; ABC transporter-like [59,60]; Homeodomain-like [61,62]; Prohibitin [63]; Initiation factor eIF-4 gamma, MA3 [64]; and HD-ZIP protein, N-terminal [65].

Further investigations are, thus, necessary to determine which of these candidate gene/protein families are directly involved in yam sex determination and cross-compatibility. This gene profiling will be useful in identifying candidate genes that can be targeted for further validation in the attempt to control flower sex and cross-pollination in yam breeding programs.

## 4. Materials and Methods

### 4.1. Plant Materials and Breeding Sites

Plant materials used for GWAS analyses consisted of 74 *D. alata* clones (33 females and 41 males) involved in hybridization activities at the International Institute of Tropical Agriculture (IITA), Nigeria, from 2010–2020. These clones included breeding lines and local landraces (Table 1). It is noteworthy that these clones were part of the 100 water yam genotypes sequenced and presented in Gatarira et al. [27] and Agre et al. [28]. The IITA yam crossing blocks in Nigeria are established at Ibadan (7°29′ N and 3°54′ E) and Abuja (9°10′ N and 7°21′ E). In Nigeria, yam fields are planted in April–May, and the harvest occurs in December–January. Soil and weather conditions at these breeding sites are presented in Appendix A.

### 4.2. Phenotypic Data Collection

#### 4.2.1. Flower Sex Phenotyping

Flower sex phenotype was assessed visually at flowering. *Dioscorea alata* male and female flowers differ morphologically in shape and size, female flowers being larger than male counterparts (Figure 8). Historical data, collected from IITA crossing blocks from 2010–2020, allowed identifying yam clones’ sex phenotypes. The sex phenotype was scored as: 1 = non-flowering, 2 = male, 3 = female as described in the yam crop ontology [66]. Although *D. alata* is strictly dioecious (no monoecious cases reported), it experiences irregular/erratic flowering like other yam species, such that a genotype may flower or not in a particular year [8,11]. For convenient analyses, we only focused on genotypes with stable/regular flowering over the considered period. The sex information of genotypes used in this study is presented in Table 1.

#### 4.2.2. Genotypes’ ACR and PHC Assessment

Calculation procedures for ACR, crossability rate and PHC were adopted from Mondo et al. [22]. The average crossability rate (ACR) was calculated using 2010–2020 historical data from IITA yam crossing blocks at Ibadan and Abuja stations, Nigeria. The ACR consisted of dividing the sum of means of a genotype’s crossability rates by the number of cross-combinations in which the genotype was involved from 2010–2020:(1)$$\mathrm{ACR}=\frac{\sum \mathrm{Crossability}\;\mathrm{rates}}{\mathrm{Number}\;\mathrm{of}\;\mathrm{cross}\;\mathrm{combinations}}$$

In Equation (1), the crossability rate was calculated as follows:
(2)$$\mathrm{Crossability}\;\mathrm{rate}\;\left(\%\right)=\frac{\mathrm{Number}\;\mathrm{of}\;\mathrm{fruits}\;\mathrm{set}}{\mathrm{Number}\;\mathrm{of}\;\mathrm{flowers}\;\mathrm{pollinated}}\times 100$$

Percentage high crossability (PHC) for a parent was calculated as the number of times the crossability rate exceeded the species overall cross-compatibility, divided by the number of cross-combinations in which that parental genotype was involved:
(3)$$\mathrm{PHC}\;\left(\%\right)=\frac{\mathrm{Number}\;\mathrm{of}\;\mathrm{crossability}\;\mathrm{rates}\;>\mathrm{overall}\;\mathrm{species}’\;\mathrm{mean}}{\mathrm{Number}\;\mathrm{of}\;\mathrm{cross}\;\mathrm{combinations}}\times 100$$

Based on a previous report, the overall mean crossability rate for *D. alata* is 31.7% [22]. The pollination information (ACR and PHC) of genotypes used in this study is presented in Table 1. This information was summarized using a cross-tabulation function implemented in Microsoft Excel.

### 4.3. DNA Extraction, Library Construction and SNP Calling

For each genotype, we collected about one gram of fresh, healthy and young leaves from a field-grown plant and immediately placed the sample on dry ice. The leaf samples were then lyophilized and kept at ambient room temperature (~25 °C). Deoxyribonucleic acid (DNA) was extracted from lyophilized leaf samples using the cetyltrimethylammonium bromide (CTAB) protocol [67] with slight changes. The DNA quality was assessed on 0.8% agarose gel and concentration was estimated using nanodrop (Amersham Bioscience, Piscataway, NJ, USA) following the manufacturer’s directives. Subsequently, 50 µL of 50 ng/µL diluted DNA of each genotype was prepared and sent to Diversity Arrays Technology (DArT) Pty Ltd., Australia, for sequencing-based DArT genotyping using the DArT marker procedure described by Agre et al. [28]. Complexity reduction methods optimized for yam at DArT were used: PstI_ad/TaqI/HpaII_ad with TaqI restriction enzyme used to eliminate a subset of PstI–HpaII. PstI-site specific adapter was tagged with 102 different barcodes enabling encoding a plate of DNA samples to run within a single lane on an Illumina GAIIx. After the sequencing, FASTQ files generated by DArT were aligned against the newly released *D. alata* genome reference [29]. Variants (SNP markers) were called using the DArT’s proprietary software, DArTSoft, as previously described [26] and a single row format was generated. Finally, hapmap and VCF files were developed from the single row format and used for the final analysis.

#### 4.3.1. Genotypic Data Analysis

Multiple sequences were generated by the DArTSeq platform using proprietary analytical pipelines (Diversity Array Technology, Canberra, Australia) and mapped to the *D. alata* v2 reference genome [29]. This produced a raw dataset (single row format) of 22,140 SNPs that were subjected to SNP markers filtering with the following criteria: markers with low sequence depth < 5; SNP markers with missing values > 20%; minor allele frequency (MAF) < 0.05; genotype quality < 20; and heterozygosity > 50. This quality control filtering resulted in 9687 good-quality SNPs distributed across the 20 chromosomes [27].

#### 4.3.2. GWAS Analysis and Identification of Putative Genes

A compressed mixed linear model (CMLM) implemented in the GAPIT R package was used to compute associations using the mixed model y=Xb+Zu+e  [68], where y is the vector of the phenotypic observations estimated for the ACR and the PHC, X represents the SNP markers (fixed effect), Z is the random kinship (co-ancestry) matrix, b is a vector representing the estimated SNP effects, u is a vector representing random additive genetic effects, and e is the vector for random residual errors.

A co-ancestry matrix from principal component representing the possible diversity subgroup and kinship was included as covariates in the GWAS model to account for population structure and familial relatedness, respectively, to reduce spurious associations. The Manhattan plot was also generated in R/CMplot to visualize GWAS results over the entire genome [69] using the GWAS output from GAPIT. The phenotypic variation explained by the model for a trait and a particular SNP was determined using stepwise regression implemented in lme4 R package. The SNP loci with significant association with the traits were determined by adjusted *p*-value using Bonferroni correction [70].

Quantile–quantile (QQ) plots were generated by plotting the negative logarithms (−log10) of the *p*-values against their expected *p*-values to fit the appropriateness of the GWAS model with the null hypothesis of no association and to determine how well the models accounted for the population structure.

To inventory potential putative genes in the vicinity of associated SNP markers for target traits, we defined a window range of 1 Mb (upstream and downstream) and genes were searched from *D. alata* generic feature format (GFF3) of the reference genome. Public database Interpro, European Molecular Biology Laboratory—European Bioinformatics Institute (EMBL-EBI) allowed us to determine the functions of the genes associated with the different SNPs identified. A Google Scholar search allowed us to obtain more information on already known identified gene or protein families. For the sex determination, two sets of genotypes were developed, male and female, and the hapmap file of associated SNP markers was developed and viewed in Tassel 5. Proportion of heterozygosity and homozygosity level was estimated across the male and female genotypes.

## 5. Conclusions

This study showed the potential of the GWAS in identifying chromosomal regions associated with sexual reproduction in *D. alata*. There is a probable association between the sex determination, ACR and PHC, since they are all controlled by the same chromosomes. Haplotype analysis confirmed the male heterogametic sex determination system for *D. alata*. This species’ reproduction traits could be controlled by multiple genes. We identified promising SNP markers for sex determination, ACR and PHC, which could be used in marker-assisted selection in yam breeding.

## Figures and Tables

**Figure 1 plants-10-01412-f001:**
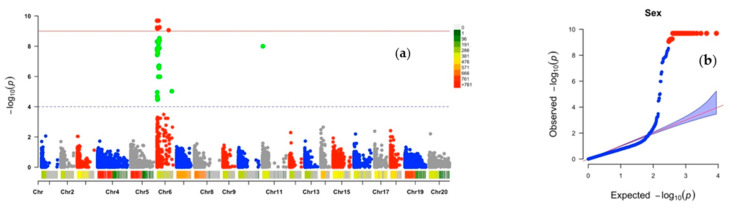
Genome-wide association analysis for plant sex determination in *D. alata*: (**a**) Manhattan plot, (**b**) quantile–quantile (Q–Q) plot. Vertical bars relate to the 20 yam chromosomes, green and red dots indicate chromosomes with influence on the target trait.

**Figure 2 plants-10-01412-f002:**
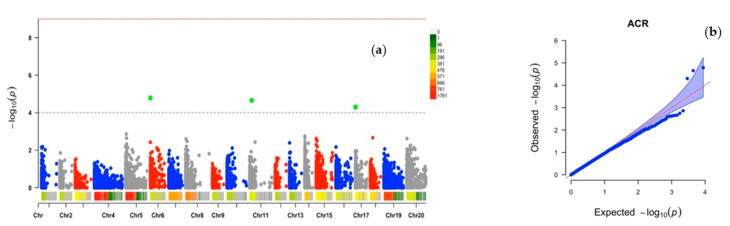
Genome-wide association analysis for average crossability rate (ACR) in *D. alata*: (**a**) Manhattan plot, (**b**) quantile–quantile (Q–Q) plot. Vertical bars relate to the 20 yam chromosomes, green dots indicate chromosomes with influence on the target trait.

**Figure 3 plants-10-01412-f003:**
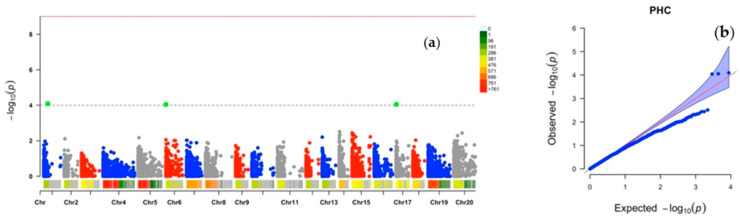
Genome-wide association analysis for percentage high crossability (PHC) in *D. alata*: (**a**) Manhattan plot, (**b**) quantile–quantile (Q–Q) plot. Vertical bars relate to the 20 yam chromosomes, green dots indicate chromosomes with influence on the target trait.

**Figure 4 plants-10-01412-f004:**
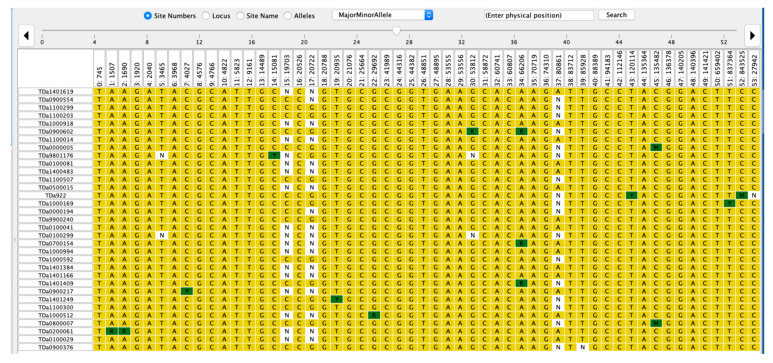
Female haplotype view for plant sex markers in *D. alata.* Yellow color refers to homozygosity while the dark green color indicates heterozygosity of the clone for the particular marker haplotype. The white color is associated with missing SNP markers information. TDa stands for Tropical *D. alata* and the number following TDa is a mention of the breeding code or accession number in the yam breeding program or the IITA Genetic Resource Center. The variable row represents the 54 sex markers identified by GWAS.

**Figure 5 plants-10-01412-f005:**
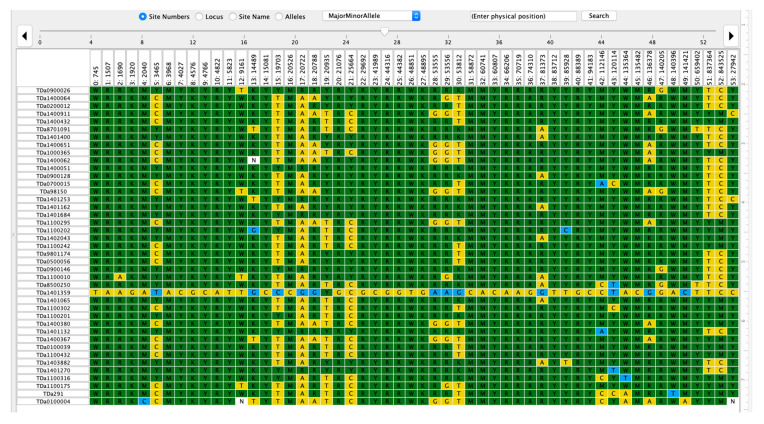
Male haplotype view for plant sex markers in *D. alata.* Yellow color refers to homozygosity while the dark green color indicates heterozygosity of the clone for the particular marker haplotype. The white color is associated with missing SNP markers information. TDa stands for Tropical *D. alata* and the number following TDa is a mention of the breeding code or accession number in the yam breeding program or the IITA Genetic Resource Center. The variable row represents the 54 sex markers identified by GWAS.

**Figure 6 plants-10-01412-f006:**
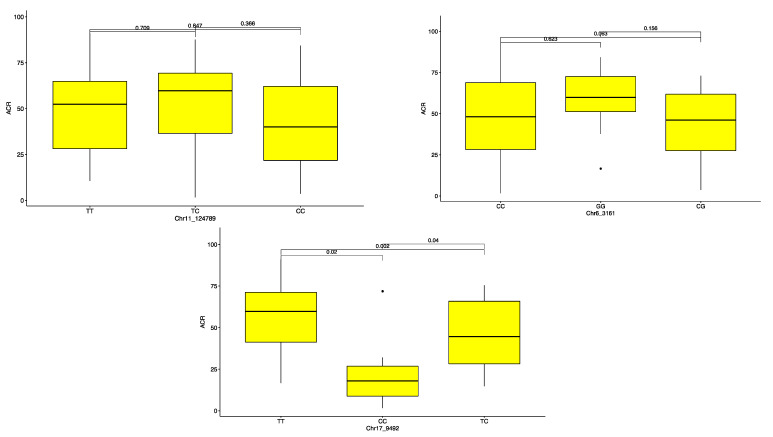
Average crossability rate (ACR) haplotype segregation in *D. alata.* The boxplots represent the segregation probabilities for each marker.

**Figure 7 plants-10-01412-f007:**
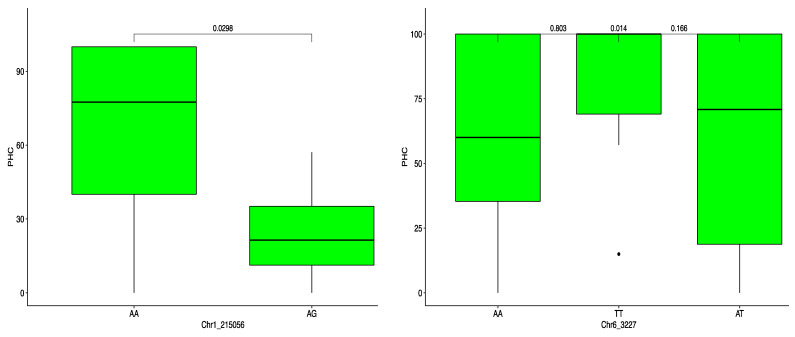
Percentage high crossability (PHC) haplotype segregation in *D. alata.* The boxplots represent the segregation probabilities for each marker.

**Figure 8 plants-10-01412-f008:**
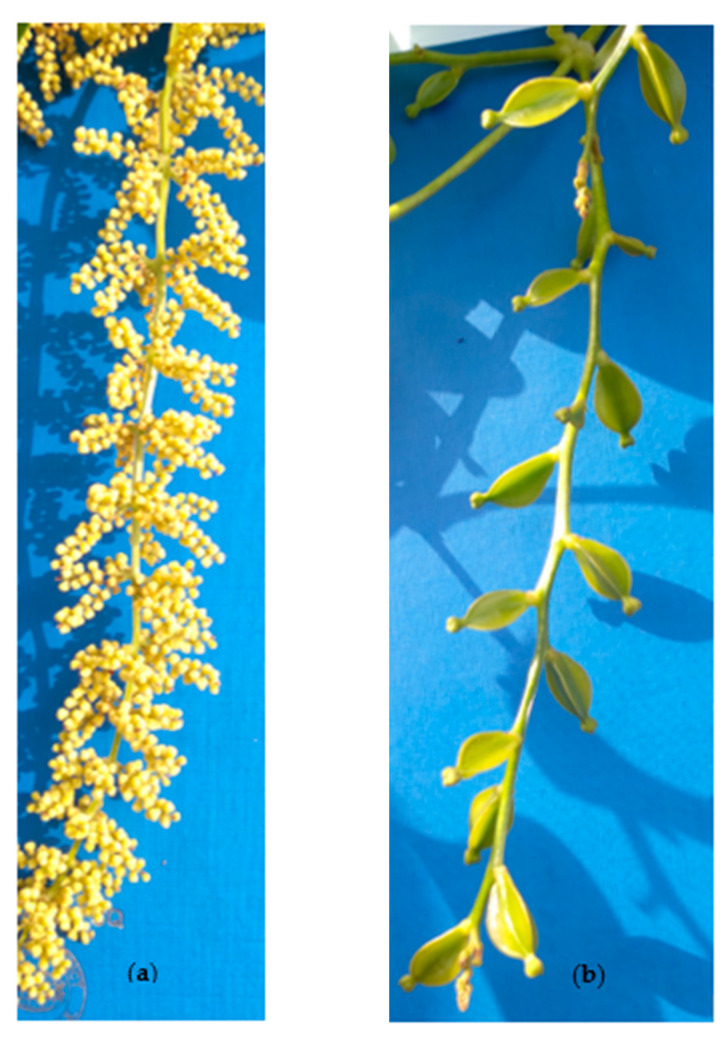
Morphological differences of *D. alata* flowers based on sex: (**a**) inflorescence with male flowers, (**b**) inflorescence with female flowers.

**Table 1 plants-10-01412-t001:** Type, sex and cross-compatibility indices (ACR and PHC) of *D. alata* clones used for GWAS analyses. Presented information is the summary of 2010–2020 historical data at IITA breeding sites, Nigeria.

Clone Name	Type	Sex	ACR (%)	PHC (%)	Cross-Combinations
TDa0000005	Breeding line	Female	28.20	38.7	38
TDa0000194	Breeding line	Female	16.64	15.0	20
TDa0100004	Breeding line	Male	25.05	25.0	11
TDa0100029	Breeding line	Female	7.74	0.0	10
TDa0100039	Breeding line	Male	37.67	63.2	19
TDa0100041	Breeding line	Female	31.19	35.3	17
TDa0100081	Breeding line	Female	41.72	57.1	21
TDa0100299	Breeding line	Female	18.41	21.4	14
TDa0200012	Breeding line	Male	27.74	35.3	40
TDa0200061	Breeding line	Female	46.80	60.0	5
TDa0500015	Breeding line	Female	42.00	66.7	21
TDa0500056	Breeding line	Male	69.12	100.0	3
TDa0700015	Breeding line	Male	73.71	100.0	3
TDa0700154	Breeding line	Female	25.98	40.0	5
TDa0800007	Breeding line	Female	36.30	42.9	7
TDa0900026	Breeding line	Male	28.51	35.7	14
TDa0900128	Breeding line	Male	87.63	100.0	3
TDa0900146	Breeding line	Male	73.21	100.0	3
TDa0900217	Breeding line	Female	32.17	47.8	23
TDa0900376	Breeding line	Female	36.76	56.5	23
TDa0900554	Breeding line	Female	42.54	80.0	5
TDa0900602	Breeding line	Female	57.14	100.0	3
TDa1000169	Breeding line	Female	69.65	100.0	3
TDa1000365	Breeding line	Male	69.29	100.0	4
TDa1000512	Breeding line	Female	66.29	71.4	7
TDa1000592	Breeding line	Female	39.75	60.0	5
TDa1000918	Breeding line	Female	49.54	57.1	7
TDa1000994	Breeding line	Female	56.21	87.5	8
TDa1100010	Breeding line	Male	23.27	27.8	18
TDa1100014	Breeding line	Female	10.56	0.0	3
TDa1100175	Breeding line	Male	84.31	100.0	3
TDa1100201	Breeding line	Male	59.82	71.4	7
TDa1100202	Breeding line	Male	76.32	100.0	3
TDa1100203	Breeding line	Female	51.71	50.0	3
TDa1100242	Breeding line	Male	69.25	100.0	3
TDa1100295	Breeding line	Male	14.77	11.1	9
TDa1100299	Breeding line	Female	63.48	80.0	5
TDa1100300	Breeding line	Female	40.04	57.1	7
TDa1100302	Breeding line	Male	77.10	100.0	3
TDa1100316	Breeding line	Male	45.25	66.7	9
TDa1100432	Breeding line	Male	59.73	100.0	6
TDa1100507	Breeding line	Female	46.12	66.7	3
TDa1400051	Breeding line	Male	64.60	100.0	3
TDa1400062	Breeding line	Male	55.51	66.7	3
TDa1400064	Breeding line	Male	59.50	85.7	7
TDa1400367	Breeding line	Male	73.04	100.0	3
TDa1400380	Breeding line	Male	30.05	0.0	3
TDa1400432	Breeding line	Male	57.37	100.0	5
TDa1400483	Breeding line	Female	82.99	100.0	4
TDa1400651	Breeding line	Male	82.91	100.0	3
TDa1400911	Breeding line	Male	58.11	100.0	3
TDa1401065	Breeding line	Male	65.52	75.0	4
TDa1401132	Breeding line	Male	70.92	100.0	7
TDa1401162	Breeding line	Male	69.12	100.0	7
TDa1401166	Breeding line	Female	68.00	100.0	3
TDa1401249	Breeding line	Female	55.24	50.0	3
TDa1401253	Breeding line	Male	91.04	100.0	3
TDa1401270	Breeding line	Male	65.00	100.0	4
TDa1401359	Breeding line	Male	90.27	100.0	3
TDa1401384	Breeding line	Female	57.50	100.0	3
TDa1401400	Breeding line	Male	56.89	80.0	5
TDa1401409	Breeding line	Female	71.83	100.0	3
TDa1401619	Breeding line	Female	20.93	20.0	5
TDa1401684	Breeding line	Male	71.74	100.0	3
TDa1402043	Breeding line	Male	75.45	100.0	3
TDa1403882	Breeding line	Male	64.33	100.0	3
TDa291	Landrace	Male	3.60	0.0	3
TDa8500250	Breeding line	Male	17.40	13.8	29
TDa8701091	Breeding line	Male	24.63	30.4	26
TDa922	Landrace	Female	12.59	0.0	7
TDa9801174	Breeding line	Male	28.24	33.3	30
TDa9801176	Breeding line	Female	1.59	0.0	13
TDa98150	Landrace	Male	21.57	11.1	18
TDa9900240	Breeding line	Female	32.43	40.0	45

ACR: average crossability rate, PHC: percentage high crossability.

**Table 2 plants-10-01412-t002:** Loci associated with sex identity, average crossability rate (ACR) and percentage high crossability (PHC) in *D. alata.* Markers are arranged in declining LOD values for plant sex and by chromosomes for ACR and PHC.

Traits	SNP Markers	Chr	Position (bp)	MAF	PVE (%)	Effect	LOD
Plant sex	Chr6_1920	6	1920	0.27	86	−1.92	9.69
	Chr6_20526	6	20,526	0.27	86	−1.92	9.69
	Chr6_21076	6	21,076	0.27	86	−1.92	9.69
	Chr6_3968	6	3968	0.27	86	−1.92	9.69
	Chr6_41989	6	41,989	0.27	86	−1.92	9.69
	Chr6_44316	6	44,316	0.27	86	−1.92	9.69
	Chr6_44382	6	44,382	0.27	86	−1.92	9.69
	Chr6_4576	6	4576	0.27	86	−1.92	9.69
	Chr6_4766	6	4766	0.27	86	−1.92	9.69
	Chr6_4822	6	4822	0.27	86	−1.92	9.69
	Chr6_48851	6	48,851	0.27	86	−1.92	9.69
	Chr6_48895	6	48,895	0.27	86	−1.92	9.69
	Chr6_5823	6	5823	0.27	86	−1.92	9.69
	Chr6_58872	6	58,872	0.27	86	−1.92	9.69
	Chr6_60741	6	60,741	0.27	86	−1.92	9.69
	Chr6_60807	6	60,807	0.27	86	−1.92	9.69
	Chr6_70719	6	70,719	0.27	86	−1.92	9.69
	Chr6_74310	6	74,310	0.27	86	−1.92	9.69
	Chr6_745	6	745	0.27	86	−1.92	9.69
	Chr6_83712	6	83,712	0.27	86	−1.92	9.69
	Chr6_88389	6	88,389	0.27	86	−1.92	9.69
	Chr6_94183	6	94,183	0.27	86	−1.92	9.69
	Chr6_140396	6	140,396	0.28	83	1.74	9.26
	Chr6_141421	6	141,421	0.28	82	1.77	9.23
	Chr6_2040	6	2040	0.28	82	1.77	9.23
	Chr6_15081	6	15,081	0.28	82	−1.82	9.23
	Chr6_4027	6	4027	0.28	82	−1.82	9.17
	Chr6_29692	6	29,692	0.28	82	−1.80	9.16
	Chr6_659402	6	659,402	0.26	81	−1.83	9.06
	Chr6_135364	6	135,364	0.26	77	1.63	8.53
	Chr6_140205	6	140,205	0.24	76	−1.75	8.40
	Chr6_135482	6	135,482	0.28	75	−1.69	8.34
	Chr6_1507	6	1507	0.28	75	−1.56	8.31
	Chr6_85928	6	85,928	0.28	74	1.56	8.15
	Chr11_27942	11	27,942	0.26	72	−1.56	7.99
	Chr6_66206	6	66,206	0.29	72	−1.59	7.89
	Chr6_136378	6	136,378	0.34	72	1.22	7.87
	Chr6_20788	6	20,788	0.34	72	1.22	7.87
	Chr6_3465	6	3465	0.43	71	1.04	7.84
	Chr6_53556	6	53,556	0.35	71	1.13	7.80
	Chr6_53555	6	53,555	0.33	71	1.24	7.79
	Chr6_1690	6	1690	0.27	70	−1.46	7.73
	Chr6_14489	6	14,489	0.29	70	1.35	7.73
	Chr6_53812	6	53,812	0.43	69	1.00	7.54
	Chr6_20722	6	20,722	0.39	68	−1.02	7.46
	Chr6_19703	6	19,703	0.41	68	−1.03	7.43
	Chr6_9161	6	9161	0.24	68	−1.48	7.42
	Chr6_120114	6	120,114	0.28	63	1.10	6.71
	Chr6_80861	6	80,861	0.40	63	−1.06	6.67
	Chr6_112146	6	112,146	0.26	62	1.18	6.55
	Chr6_837364	6	837,364	0.13	52	−1.71	5.02
	Chr6_843525	6	843,525	0.13	52	−1.53	5.02
	Chr6_20935	6	20,935	0.15	49	−1.69	4.52
	Chr6_25664	6	25,664	0.14	49	−1.55	4.47
ACR	Chr6_3161	6	3161	0.25	35	14.02	4.78
	Chr11_124789	11	124,789	0.30	33	−17.46	4.65
	Chr17_9492	17	9492	0.29	32	−20.47	4.30
PHC	Chr1_215056	1	215,056	0.03	29	−43.11	4.01
	Chr6_3227	6	3227	0.26	29	−27.36	4.04

Chr: chromosome; LOD: logarithm of the odds, MAF: minor allele frequency, PVE: phenotypic variance explained, ACR: average crossability rate, PHC: percentage high crossability.

**Table 3 plants-10-01412-t003:** Haplotype segregation for the markers associated with ACR and PHC in *D. alata*.

Trait	Marker	Haplotype	Sequence	Frequency (%)	*p*-Value	*p*-Value Adj. Signif.
ACR	Chr6_3161	Haplotype 1	CCCG	39.58	0.623	ns
		Haplotype 2	CCGG	45.14	0.063	ns
		Haplotype 3	CGGG	15.28	0.156	ns
	Chr11_124789	Haplotype 1	CCTC	26.39	0.709	ns
		Haplotype 2	CCTT	25.69	0.847	ns
		Haplotype 3	TCTT	47.92	0.366	ns
	Chr17_9492	Haplotype 1	CCTC	23.61	0.02	*
		Haplotype 2	CCTT	31.94	0.002	**
		Haplotype 3	TCTT	44.44	0.04	*
PHC	Chr1_215056	Haplotype 1	AAAG	100.00	0.03	*
	Chr6_3227	Haplotype 1	AAAT	39.58	0.803	ns
		Haplotype 2	AATT	44.44	0.014	*
		Haplotype 3	ATTT	15.97	0.166	ns

ACR: average crossability rate, PHC: percentage high crossability. ns, *,**: non-significant, significant at 5 and 1% *p*-value thresholds, respectively.

## Data Availability

Most of the data are contained within the article and Appendix A. Additional data are available on request from the corresponding author.

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
