# Peer review of "Genome-Wide Association Studies for Sex Determination and Cross-Compatibility in Water Yam (Dioscorea alata L.)"

_plants, 2021, doi:10.3390/plants10071412_

Round 1

Reviewer 1 Report

The authors in the Introduction and Methods should describe in more detail the DArT technology used.

Author Response

Reviewer 1

Comment 1. The authors in the Introduction and Methods should describe in more detail the DArT technology used.

Response 1. Many thanks for this observation and the time spent in reviewing our work. We have added the rationale for using this sequencing method in the introduction and details on the DArT technology procedure have been added to the method section.

Reviewer 2 Report

The manuscript by Mondo et al. on “Genome-wide association studies for sex determination and cross-compatibility in water yam (Dioscorea alata L.)” performed GWAS and gene annotation to identify SNPs and putative genes for sex determination, ACR and PHC. Although some information presented in the study were already available, there is novel information in the manuscript which could be applicable to breeding after validation. The following suggestions might help in improving the manuscript.

Major comments

The initial SNP calling procedure is completely missing from the manuscript. What software/procedure/criteria was used to call SNPs? Authors can contact the sequencing company to get this information and include in the manuscript.

SNP distribution across chromosomes is not uniform and are consistently highly condensed towards one end of the chromosomes. Provide explanation to this unique distribution (Cormier et al., 2019; Bredeson et al., 2021Bioarchive). This is not the case based on previous studies.

The genome annotation and gene identification section does not look connected to the rest of the manuscript. More detailed study on specific genes present in the targeted regions is required since it looks like the reference is available to authors.

First paragraph of introduction section can be shortened heavily by reducing the economic importance section. It is not necessary to present all the stats.

Minor comments

Line 44: Replace Diascorea with D. Make required changes throughout the manuscript.

Line 59: remove “controlling”

Line 119: Explain “dominant”. Is this the right word here?

Line 140: It is suggested that this sentence be written as: two markers were found … instead of “We also found …”

Line 233: The following statement is not the result obtained from this study “The candidates listed below …”. Refer to the related data or remove it from the results. It could potentially be included in the discussion with proper citations.

Line 283: Include proper citation for the statement “Our results supported previous reports …”. Since, the sex control was already reported previously, is it necessary to report this whole sub-topic in the manuscript. Highlight what is novel in the study.

Line 372: Define “stable period”

Line 384, 386 and 390: Cite the reference from where these calculations are adopted

Line 405: Confirm if it is genome sequencing or SNP genotyping?

Line 409: Explain what types of sequences “Multiple sequences”: fasta or fastq or something else?

Line 441: determination

Line 447: Delete method. Did you mean: chromosomal regions?

Line 448: Is “success” necessary?

Line 453: replace “converted for” by “used in”

Table 4: This could be part of supplementary data as no direct inference can be made from this table.

Author Response

We have attached point-by-point responses to comments and suggestions in the attached file. 

We really appreciate the time spent reviewing our paper. 

Reviewer 3 Report

Your article entitled "Genome-wide association studies for sex determination and cross-compatibility in water yam (Dioscorea alata L.)" contains worth data, interesting for a wide audience. Despite the fact that similar articles have recently been published, your research complements the data of other authors.
Therefore I find the paper very interesting and important for plant biology and worthy of publication. But it would be good to pay more attention to the differences or similarity of specific results obtained with the results obtained by other authors in similar works.

I only have one comment to the authors: in page 3 you write "74 genotypes, 33 were female and 34 were male" (line 116), but 33 plus 34 equals 67. I think this is a technical mistake.

Author Response

Reviewer 3

Comment 1. Your article entitled "Genome-wide association studies for sex determination and cross-compatibility in water yam (Dioscorea alata L.)" contains worth data, interesting for a wide audience. Despite the fact that similar articles have recently been published, your research complements the data of other authors. Therefore I find the paper very interesting and important for plant biology and worthy of publication. But it would be good to pay more attention to the differences or similarity of specific results obtained with the results obtained by other authors in similar works.

Response 1. Thanks for your good words, the differences and similarities in results have been discussed wherever needed.

Comment 2. I only have one comment to the authors: in page 3 you write "74 genotypes, 33 were female and 34 were male" (line 116), but 33 plus 34 equals 67. I think this is a technical mistake.

Response 2. Yes, it is an error from our side. We used 33 females and 41 males as stated in the materials and methods. The error has been corrected.

Round 2

Reviewer 2 Report

I would like to thank the authors for revising the manuscript. Below are a couple of points to consider which are minor at this point:

  1. What pipeline was used to call variants?
  2. Although the study included variants using GBS technology and it might matter less to get all the chromosomes covered, it is highly recommended that the authors perform variant filtering step in more lenient manner to call snps in the other end of the chromosomes. With the current map, it seems more like technical factor (being more stringent or other reasons) in the inability to find comparative number of variants in the other end of the chromosomes.

Line 45: Reference 43. Order the references in the order it is cited in the manuscript. Follow the journal’s instruction.

Author Response

Dear Reviewer,

We really appreciate quality suggestions in order to improve our manuscript.

Comment 1: What pipeline was used to call variants

Response: SNPs were called using the DArT’s proprietary software, DArTSoft, described by Jacoud et al.,  and cited in the manuscript. We have added the part in the methodology section.

Comment 2: Although the study included variants using GBS technology and it might matter less to get all the chromosomes covered, it is highly recommended that the authors perform variant filtering step in more lenient manner to call snps in the other end of the chromosomes. With the current map, it seems more like technical factor (being more stringent or other reasons) in the inability to find comparative number of variants in the other end of the chromosomes.

Response: We really appreciate the reviewer for the suggestion. We have discussed with DArT and this was what we could get from them as the alignment was done by the company. However, to solve this issue, we have sent the same materials for whole-genome re-sequencing (WGRS) to get more quality reads covering the entire genome. We will validate the current result upon getting the WGRS output.  

comment 3: Line 45: Reference 43. Order the references in the order it is cited in the manuscript. Follow the journal’s instruction.

Response: We have corrected this and it reflects in the current version. Journal instruction has been properly followed.